# Gas and Stars in the Teacup Quasar Looking with the 6-m Telescope

Alexei V. Moiseev [1,*] and Alina I. Ikhsanova [2]

1 Special Astrophysical Observatory, Russian Academy of Sciences, 369167 Nizhny Arkhyz, Russia
2 Dipartimento di Fisica e Astronomia "G. Galilei", Università di Padova, Vicolo dell'Osservatorio 3, 35122 Padova, Italy
* Correspondence: moisav@gmail.com

**Abstract:** New results on the radio-quiet type 2 quasar, known as the Teacup galaxy (SDSSJ1430+1339), based on the long-slit and 3D spectroscopic data obtained at the Russian 6-m telescope, are presented. The ionized gas giant nebula, which extends up to $r = 56$ kpc in the [O III] emission line, was mapped with the scanning Fabry–Perot interferometer. The direct estimation of the emission line ratios confirmed that the giant nebula is ionized by the AGN. Stars in the inner $r < 5$ kpc are significantly younger than the outer host galaxy and have a solar metallicity. The central starburst age ($\sim$1 Gyr) agrees with possible ages for the galactic merger events and the previous episode of the quasar outflow produced two symmetric arcs visible in the [O III] emission at the distances $r = 50$–$55$ kpc. The ionized gas velocity field can be fitted by the model of a circular rotating disk significantly inclined or even polar to the stellar host galaxy.

**Keywords:** interstellar medium (ISM); nebulae; galaxies; individual; teacup

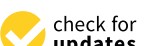



## 1. Introduction

Extended Emission-Line Regions (EELRs) detected in the outskirts of some active galaxies are considered as a result of Active Galactic Nuclei (AGN) feedback in the radiative and kinetic (jet and wind outflows) forms. The Seyfert galaxy Mrk 6 gives us a nearby, recent, and a very typical example of EELR related with AGN ionization cone observed outside of stellar host up to a projected distance 40 kpc [1]. A study of such structures allows us to better understand both the history of nuclear radiative output on the scales 0.01–0.1 Myr and the distribution of intergalactic medium (and references therein [2–4]).

One of the largest EELR among low-redshift radio-quiet AGN was recently found around the Teacup galaxy (SDSS J143029.88+133912.0). The galaxy was discovered by volunteers of the Galaxy Zoo project, and its EELR was confirmed in follow-up spectroscopic observations [5]. This type-2 quasar was nicknamed due to the morphology of the ionized gas bubbles that extend 10 kpc from the galactic center in the shape of a 'handle'. Keel et al. [6] considered Teacup as a fading AGN, whereas X-ray data have indicated that it is possible that no fading is required [7].

The Hubble Space Telescope (HST) imaging shows that the Teacup is a bulge-dominated galaxy, with a shell-like structure and a tidal tail, which have been interpreted as an indicative of merger 1–2 Gyr ago with a cold disk system and 1/10 mass ratio [8]. Another interesting feature of this system is the presence of a giant outflow generated by small-scale radio jets and/or quasar winds (Keel et al. [6], Harrison et al. [9]). This outflow appears to be responsible for the bubble-like morphology of the galaxy. The ionized gas kinematics of the outflow in the inner 15 kpc were studied several times using long-slit and 3D-spectroscopic technique [6,10–12]. However, the stellar kinematics of the host galaxy is still unknown.

The long-slit Gran Telescopio Canarias (GTC) optical spectroscopy by Villar-Martín et al. [13] reveals that the ioniziedied gas around Teacup extends up to 50 kpc in the Hα

emission line. This giant nebula has been considered by the authors as reservoir of the circumgalactic medium populated by tidal debris produced by galactic merger events. The external gas is most likely photoionized by the nuclear radiation and is significantly dynamical cold comparing with the inner 10–20 kpc region. The recent GTC deep image in the H$\alpha$ emission line clearly demonstrates that the giant Teacup EELR (155 × 87 kpc in a total size) elongates in the same direction with the main axis of the inner bubble and radio jet [14]. Several arcs and emission knots are visible up to ∼56 kpc from the AGN. However, details of gas rotation pattern are still unknown (it has been proposed that the gas maybe settled in a giant rotating disk [13]).

The giant nebula is most likely photoionized by the nuclear radiation, however to construct their diagnostic diagrams, Villar-Martín et al. [13] accepted the $I([O\,III]\,\lambda\,5007)/I(H\beta)$ flux ratio similar with the value obtained early for the internal regions, due to the lack of a green lines in the GTC spectrum.

In this work we present the results of new long-slit and 3D spectroscopic observations at the 6-m telescope of the Special Astrophysical Observatory of the Russian Academy of Sciences (SAO RAS) performed to solve the above-mentioned puzzles of the Teacup galaxy: the properties of the host stellar population and the structure and the velocity field of the extended nebula, including direct estimation of its ionization state. Following Ref. [13], we accepted Teacup redshift $z = 0.085$, which corresponds to the distance 360 Mpc ($H_0 = 71$ kms$^{-1}$ Mpc$^{-1}$) and a spatial scale 1.58 kpc arcsec$^{-1}$.

## 2. Observations and Data Analysis

The observations were carried out at the prime focus of the SAO RAS 6 m telescope with the SCORPIO-2 multi-mode focal reducer [15]. The detector was CCD $2K \times 4.5K$ E2V 42-90 . The observations in the long-slit mode were performed with the $6' \times 1''$ slit and the scale $0.36''/px$, other parameters are given in the Table 1 ($T_{exp}$—total exposure, $\theta$—mean seeing value, $\Delta\lambda$ and $\delta\lambda$ are the spectral range and resolution). The position angle $PA = 60°$ corresponds to the major axis of the nebula (Figure 1). The initial data reduction was performed in a standard way, as described in our previous papers, e.g., Ref. [16].

The 3D-spectroscopy mapping in the [O III]$\lambda$5007 emission line was carried out in the scanning Fabry–Perot interferometer (FPI) mode of SCORPIO-2 with the same low-resolution FPI that is usually used for the tunable filter imaging in the MaNGaL device [17]. During the observations we subsequently obtained a narrow band (the bandwidth ∼13 Å) images with different central wavelength: five frames spanned the spectral range around the redshifted [O III]$\lambda$5007 line with the step 5.9 Å and one frame in continuum at the central wavelength shifted on 24 Å from this emission line. The field of view at the new detector CCD $2K \times 4K$ E2V 261-80 was $6.8'$ sampled with a pixel scale $0.78''$ and $0.39''$ at the nights 19 April 2020 and 24 April, respectively; other parameters are given in Table 1.

The preliminary data reduction (bias, flat-field correction, combining individual 90 s exposures with cosmic-ray hits cleaning) was performed with IDL-based software as described in Moiseev et al. [17] for the MaNGaL scanning mode. The air-glow lines subtraction, photometric correction and phase-shift wavelength calibration were done with our software for SCORPIO-2/FPI data reduction (see description and references in Ref. [18]). The data obtained during two nights were merged in the single data cube containing low-resolution [O III] spectra in each pixel with the size $0.78''$. The total exposure was about 2.4 h. The surface brightness of the most faint detected emission filaments is about $2 \times 10^{-18}$ erg s$^{-1}$ cm$^{-2}$ arcsec$^{-2}$.

**Table 1.** Log of SCORPIO-2/6 m telescope observations.

| Mode | Date of Obs. | $T_{exp}$, s | $\theta$, '' | $\Delta\lambda$, Å | $\delta\lambda$, Å |
|---|---|---|---|---|---|
| Long-slit | 11 February 2018 | 4 × 1200 | 2.2 | 3500–7220 | 5 |
| FPI | 19 April 2020 | 41 × 90 | 3.2 | 5410–5470 | 13 |
| FPI | 24 April 2020 | 50 × 90 | 1.6 | 5410–5470 | 13 |

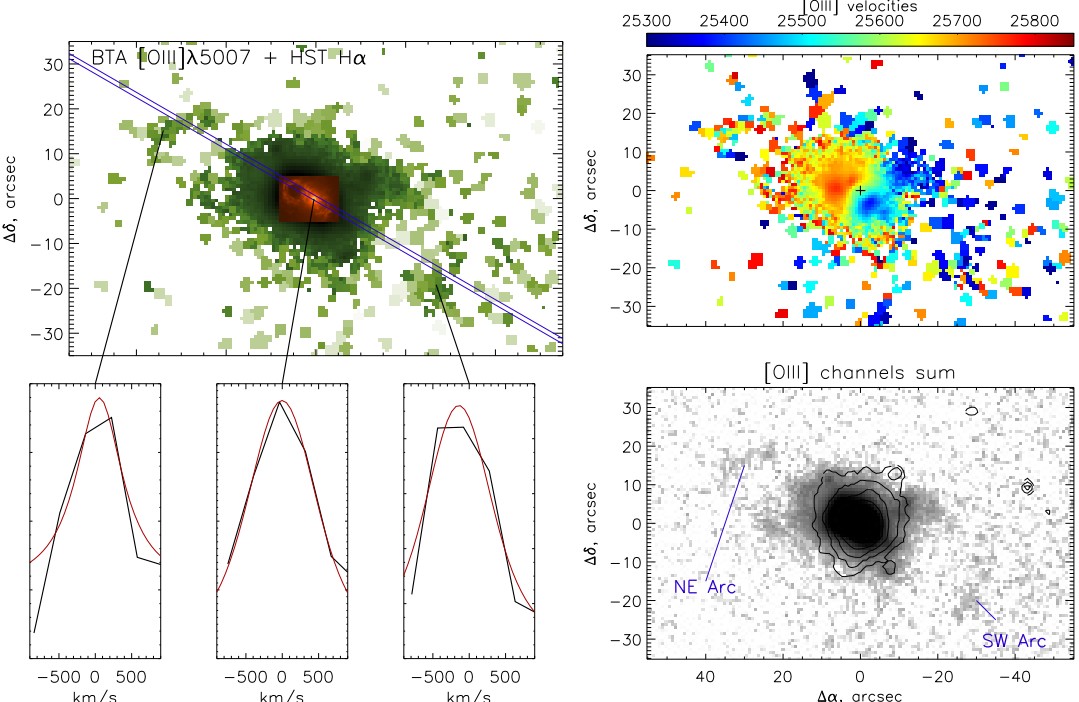

**Figure 1.** The left panel: the [O III] emission map taken with FPI after Voronoi tessellation binning (the green scale) combined with ACS HST Hα image [8] of the central bubble region (in the red scale). The purple lines mark the position of the spectrograph slit (1″ in the width). The bottom plots show the emission-line profiles in the selected regions, the Voigt fitting is marked by the red. The right panels display the ionized gas velocity field (right top) and the sum of three channels centered at the emission line in the original 0.78″ pixel scale with the isophotes of the image in the channel correspond to the stellar continuum. The most distance emission filaments are marked: the NE Arc (firstly mapped in Ref. [14]) and SW Arc.

In order to improve signal-to-noise (S/N) ratio in the spectra of the faint outer region the Voronoi tessellation was used [19]. The emission line in the binned data cube was fitted with Voigt function, which provides a good approximation of the observed FPI spectra [20]. The example of emission line profiles together with the [O III] flux maps (in the binned and original resolution) and line-of-sight velocity field is shown in Figure 1.

## 3. The Stellar Population Properties and Kinematics

The observed long-slit spectra contain a combination of the ionized gas emission and stellar absorption lines. We used the penalized pixel-fitting (pPXF) method to fit a stellar population spectrum [21], using MILES stellar spectral library by Vazdekis et al. [22] covering the range 3525–7500 Å with a twice higher spectral resolution than in the SCORPIO-2 data. In order to improve the signal-to-noise ratio, the spectra used for the pPXF analysis were binned along the slit using the step exponentially increasing with radius: from 2 px (0.7″) bins in the nucleus up to 10 px (3.6″) at the distances $r \sim 10″$ from the center. An example of the observed Teacup spectrum fitted by pPXF model is shown in Figure 2.

Figure 3 shows the distribution along the spectrograph's slit of the stellar line-of-sight velocities, age, and metallicity derived from the description of the SCORPIO-2 spectra by the single stellar population (SSP) pPXF model. This plot demonstrates that the inner $r < 3–4″$ (5–6.5 kpc) region is decoupled both in kinematic and stellar population properties from the outer host galaxy. The luminosity-averaged age of stars in the central region is significantly younger compared with the outer part of the galaxy ($\sim$1 vs. 7–9 Gyr) and has a near solar metallicity typical for the recent burst of star formation, whereas [M/H] $\approx -1$ in the outer region, which is typical for the "red-sequence" early-type galaxy.

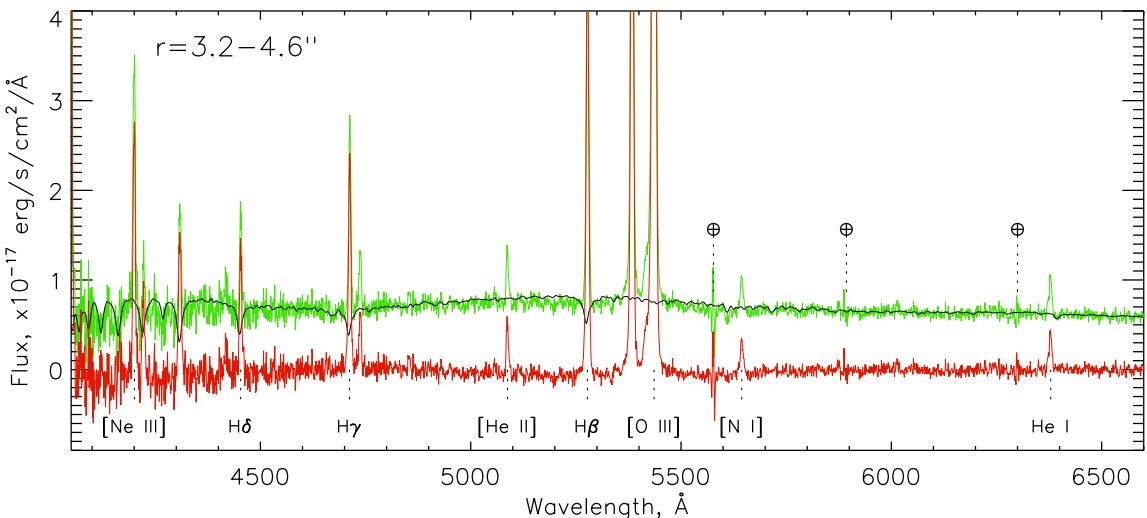

**Figure 2.** pPXF analysis of the observed spectrum (the green line) in the radial bin corresponds to the distance from the Teacup nucleus *r* = 3.2–4.6″. The stellar population model is shown in black, the red spectrum is a residual after model subtraction. The main ionized gas emission lines and position of the airglow lines are labeled.

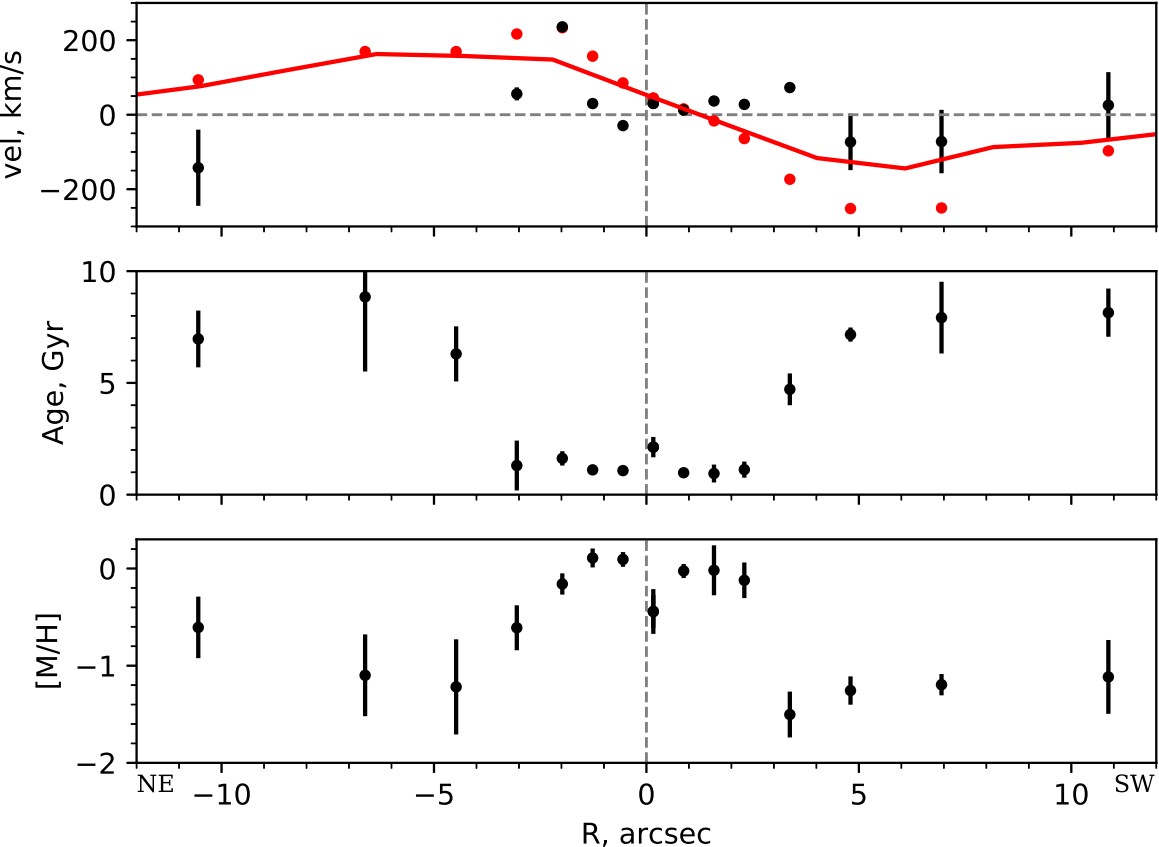

**Figure 3.** Gas/stars kinematics and stellar population properties derived from SCORPIO-2 observations at *PA* = 60°. From top to bottom: line-of-sight velocity distribution for stars (black dots) and ionized gas (red), the red line shows the velocities in this direction according to FPI data, the systemic velocity $25,482\,\mathrm{km\,s^{-1}}$ was subtracted; radial distribution of the SSP age; the same for the stellar metallicity.

The flux and velocity of the ionized gas emission lines were derived via Gaussian fitting of galactic spectra after subtraction of the pPXF model in each radial bin. During this procedure we accepted the same line-of-sight velocity separately for the system of the forbidden ([O III], [N II], [S II]) and the Balmer hydrogen lines. The top panel in the Figure 3 demonstrates an agreement between velocities of the forbidden lines in the SCORPIO-2 long-slit spectrum (red points) and a pseudo-slit cut through the FPI velocity field (red line) if the differences in the spatial and spectral resolutions will be taken into account.

According to our Gaussian fitting results, the mean flux ratio of the brightest Balmer lines at $r < 12''$ $I(H\alpha)/I(H\beta)$= 2.9 ± 0.1 is in a good agreement with the 'standard' value of an intrinsic Balmer decrement both for AGN and H II-regions (∼3.1 and 2.86, see Groves et al. [23] for a review). For this reason, we did not correct the observed spectrum for interstellar dust extinction. The reddening map presented in Gagne et al. [11] reveals that the significant extinction $E(B - V) > 0.2$ (possibly related to a circumnuclear dust lane) is detected in the inner $r < 2''$. This region is poorly resolved in our long-slit data, with a seeing value of 2.2$''$.

The kinematics of the stars significantly differs from the gaseous component: in the inner $r < 5''$ the radial gradient of stellar line-of-sight velocities is near zero with a hint of slow counter-rotation in the SW part of the velocity distribution. Such a kinematic feature is indicative of a multi-spin galaxy where the gas and stars rotate in different planes and/or in different directions. Indeed, an absence of a velocity gradient should be observed if we put a slit along the rotating disk minor axis, or if an unresolved counter-rotating stellar component presents in the kinematically distinct core [24,25]. Both features are unsurprising after galactic merging or accretion of the external gas by early type host galaxy [26].

## 4. The Gas Ionization

To discriminate the gas excitation mechanism the BPT (after [27]) diagnostic diagrams of the emission line flux ratios were widely used. In a previous work, Ref. [13] argued that even the very external regions of the Teacup giant nebula is illuminated by the AGN ionized radiation. However, the GTC spectra did not cover the green region. This is the reason why in Ref. [13] a very important $I([O III] \lambda 5007)/I(H\beta)$ line ratio for the giant EELR was accepted as the same nuclear value.

In our 6-m telescope spectrum we were able to detect the [O III] line in the emission arcs up to $r \approx 55$ kpc (Figure 4, left). The flux in the H$\beta$ was also estimated in the integrated spectrum on distances $r$ = 12–26$''$ (19–41 kpc). This gives a direct estimation of the $I([O III] \lambda 5007)/I(H\beta)$= 0.80 ± 0.08, which agrees in the errors with the prediction in Ref. [13]. The diagram $I([O III] \lambda 5007)/I(H\beta)$ vs $I([N II])/I(H\alpha)$ (Figure 4, right) clearly demonstrates that the observed line ratios both for the central and external parts of the giant nebula correspond to AGN-type ionizaion. Here we used $I([O III] \lambda 5007)/I(H\beta)$ and $I([N II])/I(H\alpha)$ ratios derived from the 6 m and GTC observations, respectively, and accepted that the $I([N II])/I(H\alpha)$ ratio in the NE region $PA = 60°$ at $r = 12 - 26''$ is the same as in the integrated GTC spectral measurements in two apertures at $r = 11.8 \pm 2.4''$ and 16.6 ± 1.8$''$, as according to Villar-Martín et al. [13].

A similar situation in other BPT-diagrams is presented by Villar-Martín et al. [13]: the line ratios of the emission lines in the Teacup EELR clearly corresponds to the ionization by UV-radiation of the central QSO.

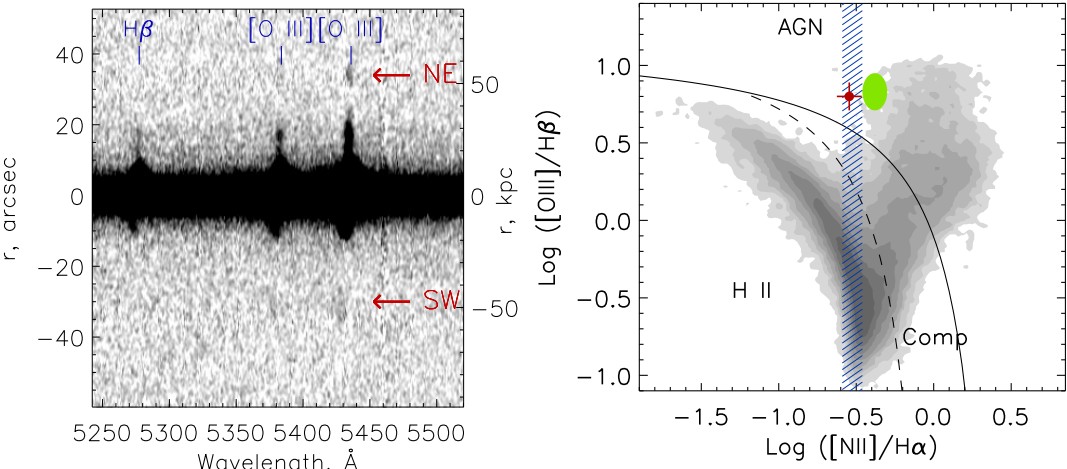

**Figure 4.** Ionized gas in the Teacup giant EELR. **Left**: a fragment of the 2D SCORPIO-2 spectrum along $PA = 60°$ around the $H\beta + [O\ III]\lambda\lambda 4959, 5007$ region. The red arrows mark the NE and WS Arcs. **Right**: a BPT diagram similar to one the presented in Figure 3 [13]. The gray contours represent the density distribution of the SDSS galaxies line ratios based on the data from Kewley et al. [28]. The division lines between the star-forming galaxies, AGN, and composite nucleus are taken from Kewley et al. [29], Kauffmann et al. [30]. The vertical blue shadowed area represents the range of values measured for the giant nebula at GTC [13]. The green filled ellipse shows the line ratios measured by Gagne et al. [11] for the areanearest the AGN regions ($r < 15$ kpc). The red point corresponds to both our SCOPRIO-2 ($I([O\ III]\ \lambda 5007)/I(H\beta)$) and GTC ($I([N\ II])/I(H\alpha)$) measurements for the distances $r = 19$–41 kpc.

## 5. The Extended Nebula: Morphology and Kinematics

At the first glance, the distribution of the ionized gas derived in SCORPIO-2 observations looks similar to the deep $H\alpha$ emission line images recently published in Villar Martín et al. [14]: the whole emission structure up to $r \approx 18''$ (~28 kpc) possesses the inner bubble (~10 kpc) and distant emission patches and filaments at $r \approx 30$–$35''$ (~47–55 kpc). The giant emission nebula is elongated in the same position angle with the radio jet direction ($PA \approx 60°$ [9]). However, the image in the hydrogen recombination line $H\alpha$ reveals the characteristic Arc+Cavity structure in the NE side of the nebula, and a very faint straight filament in the SW direction. In contrast with this picture, on the image in the high excited forbidden [O III] line (Figure 1), we see the arc-like structure in the SE side as well. It seems to be symmetric with the NE arc being in agreement with Villar Martín et al. [14], who suggested that the large scale morphology of the nebula is influenced by AGN.

We tried to describe the observed line-of-sight ionized gas velocity distribution by the model of a regular circular rotation using our adaptation of a classical 'tilted-ring' technique (see Refs. [18,26] and the references therein). The mean parameters of the gaseous disk orientation were determined from the central part of the velocity field ($r < 25''$): inclination $i_0 = 43 \pm 7°$ and the position angle $PA_0 = 62 \pm 4°$. Figure 5 shows the radial variations of the main model parameters: the circular rotation velocity $V_{ROT}$ and kinematic position angle $PA_{kin}$ for the fixed values of the inclination ($i_{kin} = i_0$) and systemic velocity. The $PA_{kin}$ was also fixed for large radii ($r > 21''$) in order to avoid unstable approximation in the sparse velocity field regions.

It is possible that the rotation curve presented in Figure 5 is affected by non-circular motions, first of all in the central $r < 15$ kpc (location of the giant bubble—'teacup handle'). This fact is indicated by both the $PA_{kin}$ radial changes and maps of the residual velocities (Figure 5, right) with well-ordered patches of negative and positive values. The maximal value of the deviations from the mean rotation ($\pm 70\,\mathrm{km\,s^{-1}}$) could be considered as the lower limit on the speed of a large-scale outflow; the projection effect can increase this value. Note that the amplitude of these non-circular motions along line-of-sight is significantly smaller than $V_{ROT}$ at the corresponding distances.

The spread of velocity measurements in the faint distant regions prevents us from interpretation of its velocity residuals. Nevertheless, we can conclude that the gas in both the NE and SW arcs lies on the common flat rotation curve, with an amplitude of 100–140 km s$^{-1}$.

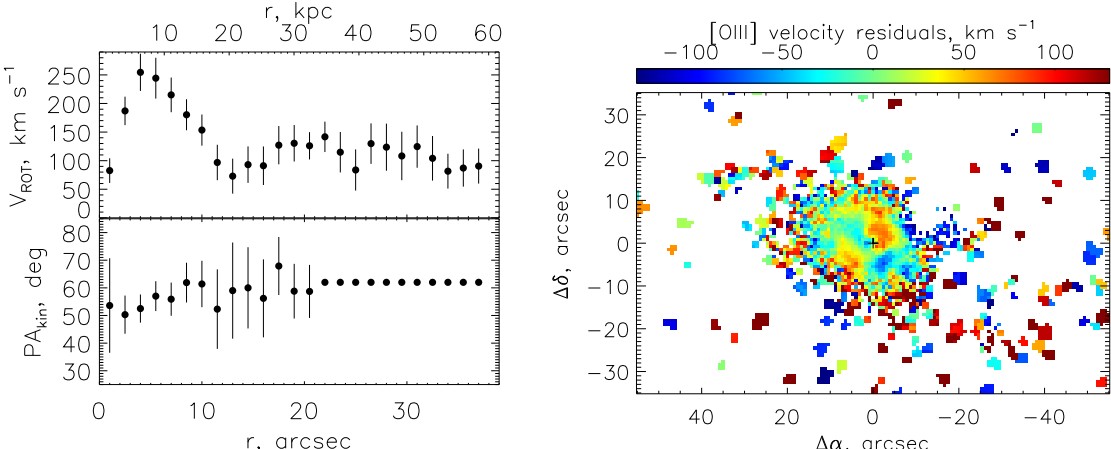

**Figure 5.** Radial variations of the tilted-ring model parameters (**left panel**): the circular rotation velocity (**top**) and the kinematic position angle (**bottom**). The **right panel** shows residual velocity field after subtraction of the disk circular rotation model.

## 6. Discussion

There is no doubt that the stellar morphology of the Teacup galaxy reveals the footprint of a merger event (Section 1). The observed shell-like narrow tidal features could be either a result of a minor merger (mass ratio ∼1/10) of a cold disk system with a massive elliptical galaxy [31,32], or a moderate merger of two disk galaxies (mass ratio ∼1/2, [33]). In both cases, the lifetime of the observed stellar arcs could be about 1–2 Gyr.

In Section 3 we find out that the stellar population in the center of the galaxy ($r < 5$ kpc) is significantly younger (SSP age ∼ 1 Gyr) and richer in metal ([M/H] ≈ 0) compared to the outer host (SSP age ∼ 7 Gyr, [M/H] ≈ −1). The age of the central starburst puts it in the above range of the merger age. This indicates that the star formation is possible triggered by interaction, as has been predicted in numerous simulations (Ref. [33] and references therein). It appears that the same interaction event fed the AGN. In this case, we do not exclude that the central star formations were partly a result of AGN outflow positive feedback. For instance, Ref. [8] found several candidates for young star clusters in the inner 8 kpc region, outside of the ionization cone boundaries.

The ionized gas distribution agrees with the picture suggested by Ref. [14] for this and other optically selected Type-2 quasars: in the systems undergoing interaction or merging, the ionized gas was spread over large spatial scales (10–90 kpc) and was illuminated by the AGN. Using the long-slit spectroscopy we directly measured the indicative line ratio $I([O\,III]\,\lambda\,5007)/I(H\beta)$ in the external regions (19–41 kpc) of the Teacup nebula for the first time. Our result fully confirmed the previous suggestion about domination of the AGN ionization, even on these distances [13].

The analysis of the [O III] velocity field demonstrates that the gas kinematics can be described in the model of a global rotating disk with the line-of-nodes major axis $PA_0 = 62 \pm 4°$. This value is significantly different from the orientation of the stellar continuum isophotes (Figure 1): $PA_* = 157 \pm 3°$ according to our estimation at $r = 18$–$20''$. The same orientation of the red continuum isophotes also appears in Figure A.16 in Ref. [22]. It implies that the gaseous disk is significantly inclined or even orthogonal to the stellar one, which also agrees with kinematically decoupling stars/gas according to long-slit data (Section 3): the stellar radial velocity gradient is near zero and significantly smaller than gaseous one, because the slit crosses the stellar host near its minor axis.

The domination of a regular circular rotation in the gaseous kinematics together with the multi-spin gas/stars configuration might constrain the parameters of the interaction produced by the Teacup galaxy. It is possible that in most of the nebula we are observing not the main galaxy gas, spread by galactic interaction and outflow, but the matter accreted from a companion with a corresponding spin orientation.

The deep [O III] emission line image reveals that the arc-like faint emission structure elongated according to the radio jet direction exists not only in the NE direction, as follows from the previous Hα map by Villar Martín et al. [14], but also appears in SW at the same distance from the AGN (50–55 kpc). The shape of both the SW and NE arcs partly repeats the well-known inner emission bubbles, suggesting its possible origin: Could it be related to a previous AGN outflow?

There are different estimations of current AGN outflow velocities in the region of 10–12 kpc radio bubbles, spanning the range of $V_{out} = 50 \ldots 150 \, \mathrm{km \, s^{-1}}$ [9,11,12]. If we accepted a conservative estimation $V_{out} > 70 \, \mathrm{km \, s^{-1}}$ according to the [O III] residual velocity map (Section 5), then the dynamical age of the NE and SW arcs is <0.8 Gyr. This value seems to be in good agreement with the age of a central star formation and minor/moderate merging. Therefore, our estimation does not contradict the fact that the most distant emission arcs around the Teacup galaxy are related with the first QSO activity episode triggered by galactic interaction, which also started a central burst of star formation. Moreover, it is possible that a circumnuclear starburst also contributed to the formation of the external emission arcs via a galactic wind (see the paper by López-Cobá et al. [34] for a review and observational examples). A more deep study of the external gas ionization properties and its kinematics (including the velocity dispersion distribution) are needed to separate the possible influence of AGN and starburst-driven wind and radio jet action on a formation of the emission arcs.

## 7. Conclusions

3D spectroscopy with scanning FPI is an old but very powerful technique used to study different astrophysical objects. In this work we present new observational capabilities of the SAO RAS 6 m telescope with low-resolution FPI, which was early used by our team as a tunable filter at 1–2.5 m telescopes. The example of the giant nebula related with radio-quiet quasar known as the Teacup galaxy demonstrates that with this device we are able to map emission lines at the surface brightness level $(few) \times 10^{-18} \, \mathrm{erg \, s^{-1} \, cm^{-2} \, arcsec^{-2}}$ during 2 h of exposures and even study the gas kinematics if the amplitude of velocity changes exceeds 50–20 $\mathrm{km \, s^{-1}}$.

The Teacup galaxy has been well studied, including multiwavelength data from X-ray to radio and integral-field spectroscopy in optical and near infrared. Nevertheless, using SCORPIO-2 long-slit and 3D spectroscopy we obtained the following new results:

- The indicative line ratio $log(I([\mathrm{O \, III}] \, \lambda \, 5007)/I(\mathrm{H}\beta))$ was directly estimated for the external regions (19–41 kpc) of the Teacup nebula. The obtained value $0.80 \pm 0.08$ lies in the range 0.7–0.9 presented by Gagne et al. [11] for the nearest to the AGN region. Together with the $I([\mathrm{N \, II}])/I(\mathrm{H}\alpha)$ ratio obtained early on in the GTC observations it confirms that the domination of AGN radiation in gas ionization is in good agreement with the conclusion of the paper [13];
- The stars in the inner $r < 5$ kpc are significantly younger and richer in metal than the outer host galaxy. The starburst age (∼1 Gyr) agrees with the timescale of a merger event, as proposed by Keel et al. [8];
- The ionized gas velocity field can be described in the term of a circular rotating disk with a flat rotation curve up to distances 50–60 kpc. This disc appears to be significantly inclined or even polar to the stellar host galaxy;
- The deep map of the [O III] emission reveals two symmetric arcs in the external region of the EELR ($r = 50$–55 kpc). It might be a remnant of the previous AGN outflow (it may be in a combination with a starburst-driven galactic wind) with the age < 0.8 Gyr.

An intriguing puzzle is the alignment of the line-of-nodes of the global rotating gaseous disk and radio jet (and ionization cone) direction. Is this a coincidence or a manifestation of a more powerful AGN influence on the surrounding gas than we expected? In any case, we hope that the results presented here will be useful for further detailed simulations of the Teacup system formation, including interaction with a companion and AGN feedback.

**Author Contributions:** Conceptualization, A.V.M.; Observations and data reduction, A.V.M.; methodology, A.V.M. and A.I.I.; validation, A.V.M. and A.I.I.; investigation, A.V.M. and A.I.I.; writing—original draft preparation, A.V.M. and A.I.I.; writing—review and editing, A.V.M. and A.I.I.; visualization, A.V.M. and A.I.I.; supervision, A.V.M.; funding acquisition, A.V.M. All authors have read and agreed to the published version of the manuscript.

**Funding:** This research was funded by grant No075-15-2022-262 (13.MNPMU.21.0003) of the Ministry of Science and Higher Education of the Russian Federation.

**Institutional Review Board Statement:** Not applicable.

**Informed Consent Statement:** Not applicable.

**Data Availability Statement:** The data underlying this article will be shared on reasonable request to the corresponding author. The raw data are available in the General Observation Archive of the SAO RAS telescopes: https://www.sao.ru/oasis/cgi-bin/fetch.

**Acknowledgments:** We obtained the observed data on the unique scientific facility "Big Telescope Altazimuthal" of SAO RAS. The long-slit observations were peformed by Dmitry Oparin. We thank the anonymous referees and Cristina Ramos Almeida for their constructive comments and Aleksandrina Smirnova for her help in preparing the text. This work is dedicated to the memory of Victor Afanasiev, whose enthusiasm and work helped make these observations. Some of the data presented in this paper were obtained from the Mikulski Archive for Space Telescopes (MAST). This research made use of NASA's Astrophysics Data System and the the NASA/IPAC Extragalactic Database (NED), which is operated by the Jet Propulsion Laboratory, California Institute of Technology, under contract with the National Aeronautics and Space Administration.

**Conflicts of Interest:** The authors declare no conflict of interest.

## Abbreviations

The following abbreviations are used in this manuscript:

| | |
|---|---|
| ASC | Advanced Camera for Surveys |
| EELR | Extended Emission-Line Regions EELR |
| FPI | Fabry–Perot interferometer |
| GTC | Gran Telescopio Canarias |
| HST | Hubble Space Telescope |
| SAO RAS | Special Astrophysical Observatory of the Russian Academy of Sciences |
| SSP | Single stellar population |

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
