# Peer review of "Gas and Stars in the Teacup Quasar Looking with the 6-m Telescope"

_universe, doi:10.3390/universe9020066_

Round 1

Reviewer 1 Report

The authors presented some astrophysical results on the radio-quiet type 2 quasar SDSSJ1430+1339 also known as Teacup galaxy obtained via the 6-meter Telescope located in Russia. Despite this object has been widely studied in the wide range of the electromagnetic wave spectrum, the used methods in the paper as the SCORPIO-2 long-slit and 3D spectroscopy have estimated some astrophysical quantities for the object. The list of such main results are presented in the conclusion of the paper. However, these estimations have also been made by the other authors, for the Reader's sake, I inquire the authors include the related estimations obtained in other papers and make a comparison with them. 

Author Response

Point 1: The authors presented some astrophysical results on the radio-quiet type 2 quasar SDSSJ1430+1339 also known as Teacup galaxy obtained via the 6-meter Telescope located in Russia. Despite this object has been widely studied in the wide range of the electromagnetic wave spectrum, the used methods in the paper as the SCORPIO-2 long-slit and 3D spectroscopy have estimated some astrophysical quantities for the object. The list of such main results are presented in the conclusion of the paper. However, these estimations have also been made by the other authors, for the Reader's sake, I inquire the authors include the related estimations obtained in other papers and make a comparison with them.

Response 1: We thank the Reviewer for the comment. We add in the Conclusion direct citations on the values we discussed (the line ratios, timescale), these sentences are highlighted in red.  

Reviewer 2 Report

Referee report on the manuscript entitled "Gas and Stars in the Teacup quasar looking with the 6-m telescope"

I read with interest the manuscript and I find interesting and publishing worthy results. The authors present new FP and long-slit data taken on the so-called Teacup galaxy, They found evindence for a radial change in the stellar populations compatible with a central star-burst triggered by the interaction that lead to the ignition of the AGN activity. They also present evidence that the gas kinematics is compatible with a rotating disc, and that the line ratios are compatible with AGN ionizaiton.

My only major concern is on the interpretation. I think that the observed line ratios, the shape of the ionized arc, and the gas kinematics could be interpreted as a shock ionization induced by a galactic wind triggered by either the central AGN or the central starburst or both. This would be a straight-forward interpretation compatible with (1) the shape of the ionized structures observed NE and SW of the galaxy, (2) the kinematics, and (3) the observed line ratios (shock ionization could produce line ratios that are in principal indistinguishable of the ones produced by an AGN). There are several studies on this topic in the literature, some of them with the authors as co-authors (e.g., Lopez-Coba et al. 2018, or more recent articles of the same author). Counter examples can be find in the literature too (e.g. Husemann et al. 2010). Why the authors have discarted that interpretation for this object? This explanation is particularly appealing as the gas "rotating" pattern is aligned with the radio-jet observed in this object. This points more clearly towards a shock induced ionization than to an AGN like (although in some cases radio-jets clean the path to the ionization of the AGN, like in Lopez-Coba et al. 2017:

https://ui.adsabs.harvard.edu/abs/2017ApJ...850L..17L/abstract)

Somme additional/minor comments:

1) It would be useful to show an example of the best fitted stellar population model using pPXF to the actual data. This would allow the reader to evaluate both the quality of the long-slit spectra and the fitting itself. This helps a lot to guess the confidence on the derived stellar population parameters (age and metallicity), shown in Fig. 2 and discussed in the text.

2) How dust extinction is traated by the stellar population fitting process? Which is the dust distribution along the locations shown in Fig. 2?

3) In Fig. 3, right panel, what are the grey points, i.e., the underlying distribution of points showing the typical seagull distribution in the BPT diagram? It is not described in the caption or in the text. If they are literature data included for reference it is needed to quote their origin and why they have been included in this plot.

4) I am not a native English speaker, but I think there are some sentences that could be re-written for a better understanding:

 Line 161: "It is possible that the presented in Fig. 4 rotation curve"

        => "It is possible that the rotation curve presented in Fig. 4"

 Line 117: "The both features" => "Both features"

 Line 17: "... host up to projected distance..." => "... host up to a projected distane..."

 Line 28: "... with shell-like structure..." => "... with a shell-like structure..."

 These are just a few examples, but I think there are more along the text. A revision by a native English speaker will improves the manuscript from my point of view.

Author Response

Point 1: My only major concern is on the interpretation. I think that the observed line ratios, the shape of the ionized arc, and the gas kinematics could be interpreted as a shock ionization induced by a galactic wind triggered by either the central AGN or the central starburst or both. This would be a straight-forward interpretation compatible with (1) the shape of the ionized structures observed NE and SW of the galaxy, (2) the kinematics, and (3) the observed line ratios (shock ionization could produce line ratios that are in principal indistinguishable of the ones produced by an AGN). There are several studies on this topic in the literature, some of them with the authors as co-authors (e.g., Lopez-Coba et al. 2018, or more recent articles of the same author). Counter examples can be find in the literature too (e.g. Husemann et al. 2010). Why the authors have discarted that interpretation for this object? This explanation is particularly appealing as the gas "rotating" pattern is aligned with the radio-jet observed in this object. This points more clearly towards a shock induced ionization than to an AGN like (although in some cases radio-jets clean the path to the ionization of the AGN, like in Lopez-Coba et al. 2017: https://ui.adsabs.harvard.edu/abs/2017ApJ...850L..17L/abstract)

Response 1: We thank the Reviewer for his/her  careful reading and useful comments. Indeed, when we wrote `the quasar  outflow’ it means a possible combination of radio-mode feedback with AGN and starburst galactic wind. We cannot separate these factors without more detailed information about line ratios and gas kinematics (including velocity dispersion) in the external arcs. In any case, we suggest that these gaseous structures are created by outburst rather than tidal interaction. We add comments (highlighted in red) in Sec 6  and 7. 

 Point 2.1 :  It would be useful to show an example of the best fitted stellar population model using pPXF to the actual data. This would allow the reader to evaluate both the quality of the long-slit spectra and the fitting itself. This helps a lot to guess the confidence on the derived stellar population parameters (age and metallicity), shown in Fig. 2 and discussed in the text.

Response 2.1: Done, see the new fig 2.

Point 2.2:  How dust extinction is traated by the stellar population fitting process? Which is the dust distribution along the locations shown in Fig. 2?

Response 2.2: we add a comment with the Balmer decrement value in Sec 3

 Point 2.3:  In Fig. 3, right panel, what are the grey points, i.e., the underlying distribution of points showing the typical seagull distribution in the BPT diagram? It is not described in the caption or in the text. If they are literature data included for reference it is needed to quote their origin and why they have been included in this plot.

Response 2.3: Done, see the updated figure and its caption.

 Point 2.4: I am not a native English speaker, but I think there are some sentences that could be re-written for a better understanding:

Response 2.4: Done, we hope that the text is easier to read now

Round 2

Reviewer 1 Report

I am pleased with the revised version of the paper. Therefore, I recommend it for publication.